# Learning to Parameterize Visual Attributes for Open-set Fine-grained Retrieval

**Shijie Wang[1], Jianlong Chang[2], Haojie Li[3]\*, Zhihui Wang[1], Wanli Ouyang[4], Qi Tian[2]**

[1]International School of Information Science & Engineering, Dalian University of Technology, China

[2] Huawei Cloud & AI, China

[3]College of Computer and Engineering, Shandong University of Science and Technology, China

[4]Shanghai Artificial Intelligence Laboratory, China

## Abstract

Open-set fine-grained retrieval is an emerging challenging task that allows to retrieve unknown categories beyond the training set. The best solution for handling unknown categories is to represent them using a set of visual attributes learnt from known categories, as widely used in zero-shot learning. Though important, attribute modeling usually requires significant manual annotations and thus is labor-intensive. Therefore, it is worth to investigate how to transform retrieval models trained by image-level supervision from category semantic extraction to attribute modeling. To this end, we propose a novel Visual Attribute Parameterization Network (VAPNet) to learn visual attributes from known categories and parameterize them into the retrieval model, without the involvement of any attribute annotations. In this way, VAPNet could utilize its parameters to parse a set of visual attributes from unknown categories and precisely represent them. Technically, VAPNet explicitly attains some semantics with rich details via making use of local image patches and distills the visual attributes from these discovered semantics. Additionally, it integrates the online refinement of these visual attributes into the training process to iteratively enhance their quality. Simultaneously, VAPNet treats these attributes as supervisory signals to tune the retrieval models, thereby achieving attribute parameterization. Extensive experiments on open-set fine-grained retrieval datasets validate the superior performance of our VAPNet over existing solutions.

## 1 Introduction

Fine-grained image retrieval attempts to build a well-generalized embedding space where the visual discrepancies among categories are clearly reflected. It plays a vital role in numerous vision applications from fashion industry, *e.g.*, retrieval of different types of shoe or clothes [22; 1], to environmental conservation, *e.g.*, retrieval endangered species [6; 38; 36; 32]. However, real-world applications probably face the input of unknown categories, and the model will treat them as known ones. As a result, the retrieval performance decays, which is unbearable in real-world applications. Open-set fine-grained retrieval is thus proposed to conduct training on known categories but retrieve unknown ones during evaluation.

Facing the unknown inputs of novel categories, an intuitive way to identify them is to capture the discriminative discrepancies of unknown categories from these inputs for identifying them. However, most existing works [19; 24; 28; 20] still focus on discriminative concepts of known categories and only capture them from unknown instances, consequently making it hard to precisely identify unknown categories. Interestingly, zero-shot learning [11; 18; 47] has proven that an unknown instance can be described integrally using a variety of visual attributes, and these attributes can be

---

\*Corresponding author: hjli@sdust.edu.cn

discovered on multiple known categories. For example, the unknown birds in Fig. 1 can be represented using visual attributes discovered from seen instances, and the combination of these attributes can clearly reflect their discrepancies, thus alleviating the problem behind open-set settings. Though important, attribute modeling usually requires significant manual annotations and thus is labor-intensive. When attribute annotations are unavailable, how to transform retrieval models trained by image-level supervision from category semantic prediction to attribute modeling is worthy of investigation.

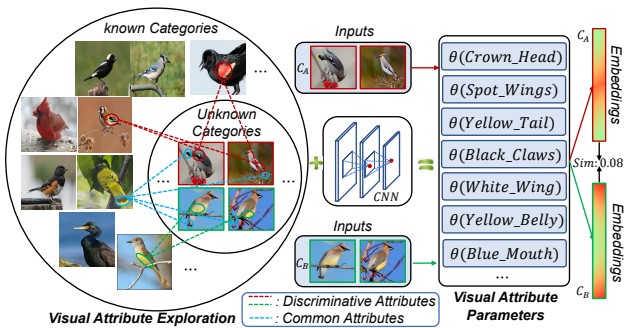

Figure 1: Motivation of the proposed VAPNet as well as the main process of retrieving unknown categories based on our visual attribute parameters. The key idea promoted throughout the paper is that the visual attributes within known categories promote the understanding of unknown inputs of novel categories. The backbone network ($CNN$) with this key idea can be transformed into a set of visual attribute parameters $\theta(\cdot)$. Therefore, given visually similar images of unknown categories ($C_A, C_B$), we feed them into our VAPNet to generate the retrieval embeddings containing various attributes, and then calculate their cosine similarity ($Sim : 0.08$) to determine whether to be distinguished. VAPNet with specific parameters transformed by visual attributes can procure in-depth semantic pattern understanding for unknown categories, improving the open-set retrieval performance eventually.

In this paper, we present a Visual Attribute Parameterization Network, termed as VAPNet, aiming to distill visual attributes from various semantics presented on seen fine-grained objects, and utilize these attributes to tune the retrieval model. Consequently, VAPNet describes the appearance of the input instances of novel categories based on its parameters tuned by visual attributes, thus transforming the retrieval model from category semantic prediction to attribute modeling. Notably, due to lacking of attribute annotations, the attributes derived from VAPNet will be not restricted to pre-defined attributes like supervised-based attribute learning works [10; 49].

Technically, VAPNet needs to parse various semantics presented in fine-grained objects, which is a prerequisite for attribute modeling. However, a feature extractor trained by image-level labels tends to focus on a few primary semantic regions (e.g., bird's head) while ignoring other visual clues (e.g. bird's body). We empirically observe that vision models can discover these overlooked object regions with rich details when taking local image patches as input compared to the whole image. Therefore, VAPNet attains some rich semantics presented in objects via parsing multiple local views randomly cropped from the input image. After that, we further apply an encoder to project these discovered semantics to a set of visual attributes. Nevertheless, due to lacking the attribute annotations, these attributes usually include some noisy patterns. To handle this limitation, we incorporate the online refinement of these attributes into the training process to iteratively improve their quality and simultaneously regard these attributes as supervision signals to tune the retrieval model, thus achieving attribute parameterization. Specifically, we design another encoder with the same structure as the above one to produce another set of visual attributes from the global features, which are used to match the visual attributes inferred by local views for providing a rich supervisory signal. To avoid optimizing two encoders instead of the retrieval model, we design the counterparts of two encoders by accumulating their parameters of all previous iterations to make supervisory signals provided by attributes tune the retrieval model directly. In this way, the features outputted by the retrieval model can be iteratively improved and fed into the optimized encoders to provide more accurate visual attributes, which, in turn, better tunes the retrieval model for visual attribute modelling.

Contributions of this paper are summarized as below:

- To the best of our knowledge, we are the first to transform the retrieval model trained by image-level supervisions from category semantic prediction into attribute modeling, thus alleviating the problem behind open-set fine-grained retrieval settings.
- We propose a novel Visual Attribution Parameterization Network, which distills visual attributes from various semantics discovered on seen fine-grained objects, and transcribes these attributes into parameters within the retrieval model, thus representing unknown categories precisely based on its parameters transformed by visual attributes.

- Extensive experiments show that open-set fine-grained retrieval task can benefit from the proposed method, and thus our VAPNet obtains significant gains of 8.6% average accuracy over recent state-of-the-art work [33] on three open-set fine-grained retrieval benchmarks.

## 2 Related Work

**Open-set fine-grained retrieval.** Existing open-set fine-grained retrieval works can be roughly divided into several groups. The first group, *localization-based scheme*, utilizes the supervision of category signals to learn discriminative embeddings [42; 52; 24; 40]. CRL [52] designs an attractive object feature extraction strategy to facilitate the retrieval task. Despite the inspiring achievement, the shortcoming of these works is that they only focus on individual samples while neglecting the inter-class and intra-class correlations between subcategories, thus reducing the retrieval performance. Therefore, the second group, *metric-based scheme*, is learning an embedding space where similar examples are attracted, and dissimilar examples are repelled [33; 41; 4; 15; 27; 51; 14; 50]. NIA [28] enforces unique translatability of samples from their respective class proxies to bring the distance of samples with the same subcategory closer. However, they still capture the discriminative details of known categories from unknown instances but neglect more details on undiscovered semantic regions, consequently impairing the retrieval performance.

Unlike the above works, FRPT [35] steers a frozen pre-trained model to perform the fine-grained retrieval task from the perspectives of sample prompting and feature adaptation. PLEor [34] could leverage pre-trained CLIP model to infer the discrepancies encompassing both pre-defined and unknown subcategories, and transfer them to the backbone network trained in the close-set scenarios. Nevertheless, it is worth noting that both of these approaches typically require more computational resources to optimize the retrieval models. This can potentially limit their practical applicability in real-world scenarios. To alleviate the problem behind open-set scenarios, we design VAPNet to explore and exploit visual attributes learnt from known instances instead of learning discriminative clues to anticipate open-set class data, improving retrieval performance in open-world scenarios accordingly.

**Visual attributes.** Attributes belong to intuitive properties of objects, which contain low-level semantics (e.g., color, texture and shape), high-level semantics (e.g., head, body and tail of objects), or even common sense (e.g., birds living on the tree) [7]. Utilizing visual attributes makes great progress on various vision tasks, including image search [17], fine-grained recognition [49; 37], scene understanding [25], and so on. Most of the previous works based on attribute learning [10; 17; 49] usually require significant manual attribute annotations and therefore is labor-intensive. Besides, the attributes learnt by these works are also restrained to pre-defined attribute labels, consequently ignoring some potentially vital information lying in visual semantics. To alleviate the aforementioned issues, recent works [43; 39] formulate an unsupervised learning strategy to project the learnt features into an attribute space. However, although the two works achieve superior performance on their corresponding vision tasks, they still are rooted in the close-set scenarios and thus make it hard to handle unknown instances. Therefore, we propose VAPNet to process more challenging scenarios, *i.e.*, open-set fine-grained retrieval tasks, by making full use of known data.

## 3 Methodology

The overall structure of VAPNet is shown in Fig. 2. It is clear that our network is mainly organized by three modules: retrieval module, attribute exploration module and attribute parameterization module. The retrieval module could extract retrieval embeddings encompassing various attributes of input objects for retrieving visually similar objects. The attribute exploration module is designed to randomly extract visual attributes from known categories. In addition, the attribute parameterization module is responsible for improving visual attributes and utilizing them as supervisory signals to tune the retrieval model.

### 3.1 Retrieval Module

The retrieval module aims at extracting basic image representations using the backbone network and producing retrieval embeddings. Thereby, the backbone network can be regarded as the retrieval model. Formally, given an image $\mathbf{X}$, let $\mathbf{F} \in \mathbb{R}^{W \times H \times C}$ be the C-dimensional with $H \times W$ feature

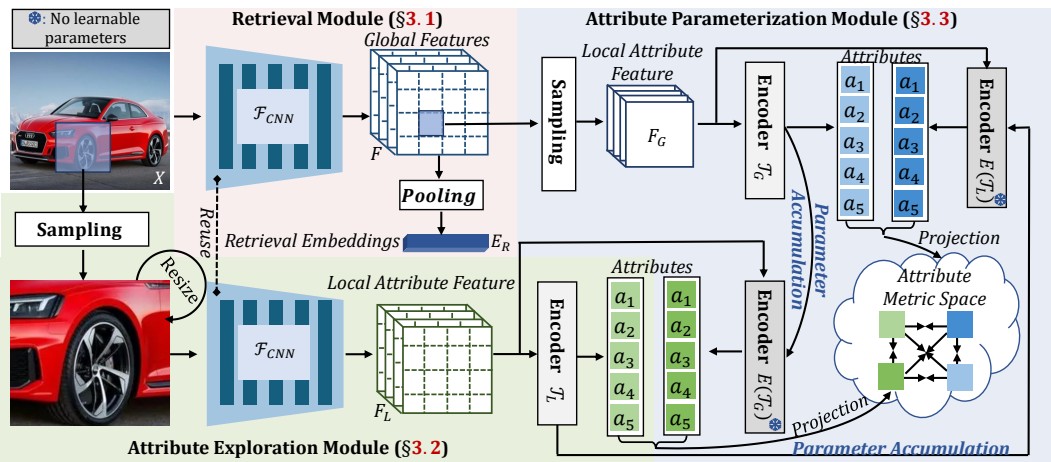

Figure 2: Detailed illustration of **visual attribute parameterization** framework. Our algorithm can be clearly divided into three main components: a retrieval module, an attribute exploration module (AEM), and an attribute parameterization module (APM). The AEM is responsible for extracting a set of visual attributes from local views, while the APM is designed to iteratively enhance the quality of these attributes. These attributes are then used as supervisory signals to fine-tune the parameters of the retrieval model. It is important to note that during testing, the single retrieval module acts as a retrieval model. The AEM and APM are only utilized during the training phase to improve the attributes and fine-tune the retrieval model parameters.

planes encoded by a backbone network $\mathbf{F} = \mathcal{F}_{\mathrm{CNN}}(\mathbf{X})$. Thus the most common way for retrieval is to embed the final feature $\mathbf{F}$ by using global average pooling operations (GAP), calculating mean values on the H × W feature plane and producing the final retrieval embeddings $\mathbf{E_R} \in \mathbb{R}^C$. It should be clarified that our VAPNet does not introduce extra computation overhead during evaluation.

## 3.2 Attribute Exploration Module

Facing the unknown input of novel categories, VAPNet aims to explore all the attributes presented in known instances as much as possible and utilize them to understand unknown categories. A non-negligible problem is that a feature extractor trained by image-level labels tends to focus on a few primary semantic regions (*e.g.*, bird's head) while ignoring other visual clues (*e.g.* bird's body). Fortunately, a feature extractor could discover object regions with rich details when replacing the input image with its local patches, as verified in Fig. 3. This suggests a proper way to focus on some overlooked object regions by making use of local image patches. Therefore, an attribute exploration module is proposed to attain some semantic clues of an input object via randomly cropping local patches from the input image. These collected semantic clues could be translated into visual attributes describing pre-defined and unknown categories.

**Input sampling.** Given an input image $\mathbf{X}$, we equally split it into n × n patches which have $3 \times \lfloor \frac{\mathrm{H}}{\mathrm{n}} \rfloor \times \lfloor \frac{\mathrm{W}}{\mathrm{n}} \rfloor$ dimensions. Here, the granularities of patches are controlled by the hyper-parameter n. In our experiments, n respectively equals to 2, 4, 8 and 16, and thus the number of patches is 340. We randomly sample M (M ≪ 340) patches from the candidate set at each iteration to construct a set of different local views $\mathbf{V} = [\mathbf{V_1}, \mathbf{V_2}, \cdots, \mathbf{V_M}]$. These local views are resized to the same scale as inputs via using bilinear interpolation. The magnification operation could directly highlight the subtle yet discriminative details in the local views, further making the backbone network more sensitive to these subtle details.

Then, the local views $\mathbf{V}$ are passed through the backbone network $\mathcal{F}_{\mathrm{CNN}}$:

$$\mathbf{F_L} = \mathcal{F}_{\mathrm{CNN}}(\mathbf{V}), \tag{1}$$

where $\mathbf{F_L} \in \mathbb{R}^{M \times C \times H \times W}$ is the local attribute feature set.

It should be clarified that the input sampling step samples local patches randomly, so it can treat all object patches equally, regardless of their discriminative ability. However, it should not be bad news for open-set fine-grained retrieval tasks, since the more diverse the visual attributes are, the

better the understanding of the unknown categories is. Furthermore, although this module picks out a few patches at each iteration, it could effectively parse the holistic structures of objects based on the accumulation of selected patches by multiple iterations.

**Attribute exploration.** In real-world applications, especially for the fine-grained tasks with small inter-class variances, attribute annotations are expensive and therefore labor-intensive. Nevertheless, unsupervised learning [26] can capture regularities in data for the purpose of extracting useful knowledge or for restoring corrupted data. Therefore, many unsupervised works [26; 2] explicitly produce internal latent units or codes from feature representation. Inspired by this, we design a visual attribute exploration to project these visual semantics attained by local views into a latent space, *i.e.*, the attribute space:

$$\mathbf{A_L^i} = \mathcal{T}_L(\mathbf{F_L^i}) = W_L \cdot g(\mathbf{F_L^i}) + b_L, \tag{2}$$

where $\mathcal{T}_L$ denotes the encoder with the weight matrix $W_L \in \mathbb{R}^{c \times k}$ and bias vector $b_L \in \mathbb{R}^k$ to process the local features produced by local views, and $g(\cdot)$ is the global average pooling operation. By the projection operation, we successfully transform the visual semantics into the visual attributes $[\mathbf{A_L^i} \in \mathbb{R}^k | i = 1, \cdots, M]$. In this way, these visual attributes could correspond to local semantic knowledge of fine-grained objects ( *e.g.*, red head, spotted wings, *etc.*), so the group of certain attributes can clearly describe an unknown category and reflect its discriminative discrepancies.

## 3.3 Attribute Parameterization Module

Visual attributes produced by local views serve as supervisory signals to tune the retrieval model, further make it be transformed from category semantic extraction to attribute modelling. Nevertheless, due to lacking the attribute annotations, these visual attributes substantively include some noisy patterns. The retrieval model supervisied by these noisy attributes is harmful to parsing fine-grained objects. To this end, we propose an attribute parameterization module to incorporate the online refinement of these attributes into the training process to iteratively improve them and simultaneously regard these attributes as supervisory signals to tune the retrieval model. In this way, the retrieval model could capture visual attributes from input instances, thus achieving attribute parameterization.

**Attribute sampling.** To match visual attributes provided by local views, we need to process the final features $\mathbf{F}$ inferred by an input image $\mathbf{X}$ to produce another group of attributes, which are used to match these visual attributes provided by local views. Concretely, forwarding $\mathbf{F}$ into a sampler extracts the corresponding local attribute feature sets $\mathbf{F_G} = [\mathbf{F_G^1}, \mathbf{F_G^2}, \cdots, \mathbf{F_G^M}]$ according to four coordinates of local views. Specifically, we utilize RoIAlign operation proposed in Mask-RCNN [8] to accurately extract the corresponding local features. Then, we project these local representation in another attribute space:

$$\mathbf{A_G^i} = \mathcal{T}_G(\mathbf{F_G^i}) = W_G \cdot g(\mathbf{F_G^i}) + b_G, \tag{3}$$

where $\mathcal{T}_G$ represents an encoder with the weight matrix $W_G \in \mathbb{R}^{c \times k}$ and bias vector $b_G \in \mathbb{R}^k$ to process the local features produced by the final features, and $\mathbf{A_G^i} \in \mathbb{R}^k$ denotes the $i$-th attribute in the attribute set $\mathbf{A_G} \in \mathbb{R}^{M \times k}$.

With these visual attributes including $\mathbf{A_L}$ and $\mathbf{A_G}$ and local features containing $\mathbf{F_L}$ and $\mathbf{F_G}$, the attribute pairs $\mathbf{A}$ and local feature pairs $\mathbf{F_P}$ can be organized as:

$$\begin{aligned} \mathbf{A} &= [(\mathbf{A_L^1}, \mathbf{A_G^1}), (\mathbf{A_L^2}, \mathbf{A_G^2}), \cdots, (\mathbf{A_L^M}, \mathbf{A_G^M})], \\ \mathbf{F_P} &= [(\mathbf{F_L^1}, \mathbf{F_G^1}), (\mathbf{F_L^2}, \mathbf{F_G^2}), \cdots, (\mathbf{F_L^M}, \mathbf{F_G^M})]. \end{aligned} \tag{4}$$

**Attribute parameterization constraint.** This constraint is responsible for improving these visual attributes and utilizing them to tune the retrieval model. Specifically, as the local attribute features fed to each encoder come from global and local views, this encoder only distills visual attributes only from its corresponding view. Thus, given local attribute features, no matter which view they come from, if two encoders provide the same attribute, it means this feature can be regarded as from both two sources. In other words, the feature discrepancy between the local attribute features inferred from both global and local views is effectively eliminated. Thereby, the above process could optimize two encoders to iteratively improve visual attributes, and utilize these attributes to modify the features inferred by local and global views, thus achieving attribute parameterization.

To this end, the attribute parameterization constraint $\mathcal{L}_c$ can be formulated as:

$$\mathcal{L}_c(\mathbf{A}, \mathbf{F_P}) = \sum_{i=1}^{M} [\mathcal{T}_G(\mathbf{F_L^i})\log\frac{\mathcal{T}_G(\mathbf{F_L^i})}{\mathbf{A_L^i}} + \mathcal{T}_L(\mathbf{F_G^i})\log\frac{\mathcal{T}_L(\mathbf{F_G^i})}{\mathbf{A_G^i}}]. \tag{5}$$

This loss encourages the two encoders to produce the same attributes for the same visual content, no matter which view it comes from, thus achieving attribute parameterization. However, training the model with Eq. (5) directly will make the attributes provided by two encoders become similar quickly since the encoders learn the attributes from another view according to Eq. (5). Therefore, using Eq. (5) more optimizes the parameters of two encoders, but has less impact on the parameters of the retrieval model.

To handle this limitation, we propose two mean encoders with the same structure as the above ones to produce attributes for features of another view. In this way, Eq. (5) can be written as

$$\hat{\mathcal{L}}_c(\mathbf{A}, \mathbf{F_P}) = \sum_{i=1}^{M} [E[\mathcal{T}_G](\mathbf{F_L^i})\log\frac{E[\mathcal{T}_G](\mathbf{F_L^i})}{\mathbf{A_L^i}} + E[\mathcal{T}_L](\mathbf{F_G^i})\log\frac{E[\mathcal{T}_L](\mathbf{F_G^i})}{\mathbf{A_G^i}}], \tag{6}$$

where $E[\mathcal{T}_G]$ and $E[\mathcal{T}_L]$ denote the mean encoders without learnable parameters, respectively. Their parameters can be updated in a temporal average manner. Concretely, at the $t$-th iteration, parameters $E[\mathcal{T}_G](\theta_G)$ and $E[\mathcal{T}_L](\theta_L)$ are accumulated by

$$\begin{aligned} E^{(t)}[\mathcal{T}_G](\theta_G) &= (1-\alpha)E^{(t-1)}[\mathcal{T}_G](\theta_G) + \alpha\theta_G, \\ E^{(t)}[\mathcal{T}_L](\theta_L) &= (1-\alpha)E^{(t-1)}[\mathcal{T}_L](\theta_L) + \alpha\theta_L, \end{aligned} \tag{7}$$

where $E^{(t)}[\mathcal{T}_G](\theta_G)$, $(E^{(t)}[\mathcal{T}_L](\theta_L))$ and $E^{(t-1)}[\mathcal{T}_G](\theta_G)$, $(E^{(t-1)}[\mathcal{T}_L](\theta_L))$ respectively denote the parameters of the mean encoders in current iteration and last iteration, and $\theta_G = (W_G, b_G)$ and $\theta_L = (W_L, b_L)$ are the learnable parameters of $\mathcal{P}_G$ and $\mathcal{P}_L$ at the current iteration, respectively. The mean encoders are initialized as $E^{(0)}[\mathcal{T}_G](\theta_G) = \theta_G$ and $E^{(0)}[\mathcal{T}_L](\theta_L) = \theta_L$. The hyper-parameter $\alpha$ is the updating ratio within the range of $[0, 1)$.

Since the two mean encoders do not introduce learnable parameters, the attribute parameterization constraint could directly penalize the retrieval model and make its parameters be adjusted through back propagation. More importantly, the mean encoders could consider the knowledge learned from all previous stages to form more robust attributes for the current stage. Therefore, they have another important property that could remain sensitive even for rare attributes, consequently staying well generalized when facing unknown categories. After the attribute parameterization operation, VAPNet will transform the retrieval model from category semantic extraction to attribute modeling, allowing the utilization of visual attributes to anticipate open-set class data.

### 3.4 Loss Functions

The retrieval model with specific parameters supervised by visual attributes can extract the attributes presented in objects and further procure in-depth semantic pattern understanding. For fine-grained understanding, these extracted visual attributes should clearly reflect the discrepancies of an objects, so that we can better identify visually similar objects. Thereby, we propose an auxiliary constraint based on the cross-entropy loss to ensure that these extracted visual attributes can contribute to decision boundary:

$$\mathcal{L}_a = y\log(C(g(\mathbf{F}))), \tag{8}$$

where $y$ denotes the ground-truth label of the corresponding input, and $C(\cdot) \in \mathbb{R}^{c \times N}$, $N$ is the number of category in the training set.

The total loss $\mathcal{L}$ of VAPNet can be formulated as:

$$\mathcal{L} = \mathcal{L}_a + \lambda\hat{\mathcal{L}}_c, \tag{9}$$

where $\lambda$ is the hyper-parameter to balance the contributions of the individual loss item.

# 4 Experiments

## 4.1 Experimental Setup

**Datasets.** CUB-200-2011 dataset [5] contains 200 bird subcategories with 11,788 images. We utilize the first 100 classes (5,864 images) in training and the rest (5,924 images) in testing. The Stanford Cars dataset [16] contains 196 car models of 16,185 images. The spilt in Stanford Cars [16] is also similar to CUB, which is split into the first 98 classes (8,054 images) for training and the remaining classes (8,131 images) for testing. FGVC Aircraft dataset [23] is divided into first 50 classes (5,000 images) for training and the rest 50 classes (5,000 images) for testing. In Shop Clothes Retrieval (In-Shop) [21] contains 7,982 subcategories with 52, 712 images, and we use the 3,997 classes (25,882 images) in training and the rest 3,985 classes in testing. In-Shop is divided between a query (14,218 images) and a gallery set (12,162 images).

**Evaluation protocols.** We evaluate the retrieval performance by *Recall@K* with cosine distance, which is average recall scores over all query images in the test set and strictly follows the setting in the pioneer work [31]. Specifically, for each query, our model returns the top $K$ similar images. In the top $K$ returning images, the score will be 1 if there exists at least one positive image, and 0 otherwise.

**Implementation Details.** For backbone network, we apply the widely-used Resnet-50 [9] in our experiments with the pre-trained parameters. The input raw images are resized to $256 \times 256$ and cropped into $224 \times 224$. We train our models using Stochastic Gradient Descent (SGD) optimizer with weight decay of 0.0001, momentum of 0.9, and batch size of 32. We adopt the commonly used data augmentation techniques, *i.e.*, random cropping, left-right flipping, and color jittering for robust feature representations. Our model is trained end-to-end on one NVIDIA 2080Ti GPUs for acceleration. The initial learning rate is set to $10^{-5}$, with exponential decay of 0.9 after every 5 epochs. The total number of training epochs is set to 200.

## 4.2 Ablation Study

The proposed VAPNet is optimized by a combination of two loss functions, an auxiliary loss $\mathcal{L}_a$ and an attribute parameterization constraint $\mathcal{L}_c$ or $\hat{\mathcal{L}}_c$, which play different roles in guiding our model to understand unknown categories. Here, we perform thorough ablation experiments on CUB-200-2011 and Stanford Cars datasets to further validate the effectiveness of each loss function. Tab. 1 shows quantitative comparisons between different combinations of loss functions. The

Table 1: Comparison of performance and efficiency on CUB-200-2011 and Stanford Cars datasets using different combinations of constraints. "R@1" denotes the Recall@1 retrieval performance. "Time" is the time of extracting retrieval embeddings.

| $\mathcal{L}_a$ | $\mathcal{L}_c$ | $\hat{\mathcal{L}}_c$ | CUB R@1 | CAR R@1 | Time |
|---|---|---|---|---|---|
| ✓ | | | 69.5% | 89.3% | 21.1ms |
| ✓ | ✓ | | 71.4% | 91.2% | |
| | | ✓ | 74.1% | 92.7% | 21.1 ms |
| ✓ | | ✓ | 76.2% | 94.8% | |

baseline method only using $\mathcal{L}_a$ obtains 69.5% and 89.3% Recall@1 accuracy on CUB-200-2011 and Stanford Cars datasets, respectively. The results reflect that the network only learns the discriminative object regions instead of the visual attributes, consequently impairing the retrieval performance of unknown categories. An addition of $\mathcal{L}_c$ improves Recall@1 from 69.5% to 71.4%. However, $\mathcal{L}_c$ optimizes the encoders to translate the semantics into visual attributes, instead of parameterizing the attributes into the retrieval model. During testing, our VAPNet still makes it hard to handle unknown categories, thus limiting the performance gains. To handle this limitation, we improve the attribute parameterization constraint to force this loss function to directly optimize the parameters within backbone network. As expected, $\hat{\mathcal{L}}_c$ can effectively make the backbone network sensitive to visual attributes and understand unknown categories accordingly. Furthermore, we also verify the effectiveness of $\mathcal{L}_a$, which can ensure that these visual attributes learnt from known categories keep discriminative. As shown in Tab. 1, the proposed VAPNet achieves 76.2% and 94.8% Recall@1 performance owing to the combination of $\hat{\mathcal{L}}_c$ and $\mathcal{L}_a$ on two widely-used benchmarks. Additionally, during testing, the retrieval embedding extraction time remains the same as that of the baseline model, as the additional attribution exploration modules (AAM and APM) are only used during training.

Table 2: Comparison of different methods on CUB-200-2011, Stanford Cars 196 and FGVC Aircraft datasets.

| Method | CUB-200-2011 | | | | Stanford Cars 196 | | | | FGVC Aircraft | | | |
|---|---|---|---|---|---|---|---|---|---|---|---|---|
| | 1 | 2 | 4 | 8 | 1 | 2 | 4 | 8 | 1 | 2 | 4 | 8 |
| SCDA [42] | 57.3 | 70.2 | 81.0 | 88.4 | 48.3 | 60.2 | 71.8 | 81.8 | 56.5 | 67.7 | 77.6 | 85.7 |
| PDDM [3] | 58.3 | 69.2 | 79.0 | 88.4 | 57.4 | 68.6 | 80.1 | 89.4 | - | - | - | - |
| CRL [52] | 62.5 | 74.2 | 82.9 | 89.7 | 57.8 | 69.1 | 78.6 | 86.6 | 61.1 | 71.6 | 80.9 | 88.2 |
| CEP [4] | 69.2 | 79.2 | 86.9 | 91.6 | 89.3 | 93.9 | 96.6 | 98.1 | - | - | - | - |
| HDCL [46] | 69.5 | 79.6 | 86.8 | 92.4 | 84.4 | 90.1 | 94.1 | 96.5 | 71.1 | 81.0 | 88.3 | 93.3 |
| DGCRL [53] | 67.9 | 79.1 | 86.2 | 91.8 | 75.9 | 83.9 | 89.7 | 94.0 | 70.1 | 79.6 | 88.0 | 93.0 |
| DCML [50] | 68.4 | 77.9 | 86.1 | 91.7 | 85.2 | 91.8 | 96.0 | 98.0 | - | - | - | - |
| DRML [51] | 68.7 | 78.6 | 86.3 | 91.6 | 86.9 | 92.1 | 95.2 | 97.4 | - | - | - | - |
| DAS [20] | 69.2 | 79.3 | 87.1 | 92.6 | 87.8 | 93.2 | 96.0 | 97.9 | - | - | - | - |
| IBC [29] | 70.3 | 80.3 | 87.6 | 92.7 | 88.1 | 93.3 | 96.2 | 98.2 | - | - | - | - |
| NIA [28] | 70.5 | 80.6 | - | - | 89.1 | 93.4 | - | - | - | - | - | - |
| Proxy [13] | 71.1 | 80.4 | 87.4 | 92.5 | 88.3 | 93.1 | 95.7 | 97.5 | - | - | - | - |
| HIST [19] | 71.4 | 81.1 | 88.1 | - | 89.6 | 93.9 | 96.4 | - | - | - | - | - |
| ETLR [14] | 72.1 | 81.3 | 87.6 | - | 89.6 | 94.0 | 96.5 | - | - | - | - | - |
| PNCA [33] | 72.2 | 82.0 | 89.2 | 93.5 | 90.1 | 94.5 | 97.0 | 98.4 | - | - | - | - |
| VAPNet | **76.2** | **84.6** | **90.1** | **94.0** | **94.8** | **96.3** | **98.0** | **98.6** | **87.2** | **91.7** | **95.0** | **96.3** |

## 4.3 Comparison with the State-of-the-Art Methods

**Open-set Fine-grained Object Retrieval.** We compare our VAPNet with some state-of-the-art approaches. In Tab. 2, the performance of different methods on CUB-200-2011, Stanford Cars-196, and FGVC Aircraft datasets is reported, respectively. In the table from top to bottom, the methods are roughly divided into three groups, *i.e.*, localization-based networks, metric-based frameworks, and our VAPNet.

As shown in Tab. 2, it is obvious that the retrieval performance obtained by our VAPNet is better than other methods no matter whether the localization-based or metric-based schemes are adopted. Concretely, existing works based on localization schemes, *i.e.*, CEP [4] and HDCL [46], tend to project the final retrieval embeddings into a category space. Despite the encouraging achievement, the shortcoming of these works is that they only focus on individual samples while neglecting the correlations among subcategories, thus limiting the retrieval performance. To address this problem, the effectiveness of these models based on metric schemes, *i.e.*, ETLR [14] and PNCA [33], can be largely attributed to their precise identification of negative/positive pairs through the manipulation of distances, which indirectly enhances the discriminative power of features. However, these existing works, *e.g.*, CEP [4], HIST [19] and PNCA [33], follow a close-set learning setting, where all the categories are pre-defined, to learn the discriminative and generalizable embeddings for identifying the visually similar objects of unknown subcategories. It is thus very challenging for a feature extractor trained in closed-set scenarios with classification or metric supervisions to capture discriminative discrepancies from unknown subcategories, consequently impairing the retrieval performance.

To handle this limitation, our VAPNet focuses on learning visual attributes instead of discriminative clues to understand the unknown categories and clearly reflect their discriminative discrepancies, thus achieving a clear improvement of state-of-the-art methods.

**Large-scale Product Retrieval.** Our VAPNet exceeds all the existing methods and achieves the best performance with a retrieval accuracy of 93.9%, as shown in Tab. 3. Besides, we beat the second-best work CEP [4] and get a better result with a relative accuracy improvement of 3.0%. By leveraging the visual attributes learned from known instances to identify category-specific discrepancies, our VAPNet demonstrates impressive generalization capabilities.

Table 3: Comparison of different state-of-the-art methods on In-shop dataset.

| method | 1 | 10 | 20 | 30 | 40 |
|---|---|---|---|---|---|
| HDC [45] | 62.1 | 84.9 | 89.0 | 91.2 | 92.3 |
| ABE [12] | 87.3 | 96.7 | 97.9 | 98.2 | 98.5 |
| EPSHN [44] | 87.8 | 95.7 | 96.8 | - | - |
| NSM [48] | 89.4 | 97.8 | 98.7 | 99.0 | - |
| MS [41] | 89.7 | 97.9 | 98.5 | 98.8 | 99.1 |
| CEP [4] | 90.6 | 98.0 | 98.6 | 98.9 | 99.1 |
| PNCA [33] | 90.9 | 98.2 | 98.9 | 99.1 | 99.4 |
| Our VAPNet | **93.9** | **98.7** | **99.1** | **99.4** | **99.6** |

## 4.4 Discussions

**Patch number $M$.** Tab. 4 ablates the role of patch number $k$ in the attribute exploration module (§3.2). The optimal value is $M = 4$ (our default). Moreover, VAPNet is robust when $M$ is in $[4, \cdots, 16]$, showing that it is beneficial

Table 4: The retrieval accuracy on CUB-200-2011 of model trained with different number $M$ of local views in §3.2.

| Number $M$ | 1 | 2 | 4 | 8 | 16 |
|---|---|---|---|---|---|
| Recall@1 | 73.6% | 74.9% | 76.2% | 76.1% | 76.2% |

to spot visual attributes in a relatively many local regions. It is worth mentioning that when $M$ is too large, the training time grows exponentially. However, when $M$ is too small, the performance degrades due to easily overlooking some undiscovered regions. The results reveal that the local regions help the model attain accurate attributes, leading to better understanding unknown categories.

**Attribute dimension $k$.** We investigate the necessity of diverse attribute dimensions $k$ for retrieval performance. As reported in Tab. 5, the dimension $k$ stores the attribute knowledge. Although the large dimension could hold more infor-

Table 5: Comparison of model trained with different dimension $k$ of visual attributes on CUB-200-2011.

| Dim $k$ | 32 | 64 | 128 | 256 | 512 |
|---|---|---|---|---|---|
| Recall@1 | 74.7% | 75.9% | 76.1% | 76.2% | 76.0% |

mation related to attributes and has less impact on retrieval performance, it is easy to contain more useless information and increase storage overhead. However, when the dimension is small, it is not enough to precisely represent visual attributes, leading to degraded performance. Therefore, the optimal dimension is $k = 256$.

**Updating ration $\alpha$.** Tab. 6 reports the accuracy of using diverse updating ratios in Eq. (7). Notably, after increasing the updating ratio, the retrieval performance reduces progressively. These results reveal that a large updating ratio

Table 6: Evaluation results on CUB-200-2011 of model trained with different updating ratio $\alpha$ in Eq. (7).

| Ratio $\alpha$ | 0.1 | 0.2 | 0.4 | 0.6 | 0.8 |
|---|---|---|---|---|---|
| Recall@1 | 75.9% | 76.2% | 75.1% | 73.9% | 72.6% |

quickly updates the projectors more relying on the learning parameters on the current stage, thus easily degrading the discrepancies between two projectors. Moreover, when using a small updating ratio, the projectors keep sensitive to previous learning knowledge and easily keep different during optimization, thus extracting precisely visual attributes from given features.

**Balanced parameter $\lambda$.** There is one balanced parameter in Eq. (10). The sensitivity analysis of the parameter are performed on CUB-200-2011 and the evaluation results are presented in Tab. 7. It is observed that the performances

Table 7: Quantitative performance of model trained with different weight $\lambda$ in loss function in Eq. (10) on CUB-200-2011.

| Weight $\lambda$ | 1 | 5 | 10 | 15 | 20 |
|---|---|---|---|---|---|
| Recall@1 | 73.8% | 75.2% | 76.2% | 75.0% | 74.9% |

of our VAPNet are not stable with the variation of $\lambda$ (from 1 to 20). The retrieval performance increases as $\lambda$ grows to 10, the consistency constraint with a large balanced parameter would force the network to focus on visual attributes. However, this constraint has a larger balanced parameter, which makes the network neglect the discriminative ability of visual attributes, thus reducing retrieval performance. The upper bound of retrieval performance may saturate at $\lambda = 10$ for learning visual attributes from known categories.

## 4.5 Visual Attribute Analysis

Interpreting visual attributes is difficult because these attributes are optimized in a latent space. We resort to an indirect way to interpret these attributes by visualizing their sources (*i.e.*, Fig. 3) to display the content within them, and the features influenced by them (*i.e.*, Fig. 4) to indirectly track these visual attributes.

Our VAPNet distills visual attributes from some local regions randomly cropped from inputs. Therefore, we provide some activation maps generated by Grad-CAM [30] to display some visual clues of interest in the attributes. In Fig. 3, these semantics provided by local regions could grab some rich details, and thus attributes projected by them could clearly represent these regions. Besides, we can also observe that the response maps of the local views highlight more object details compared to that

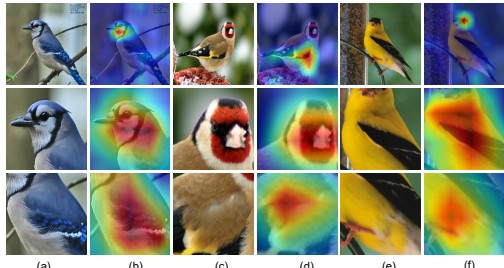 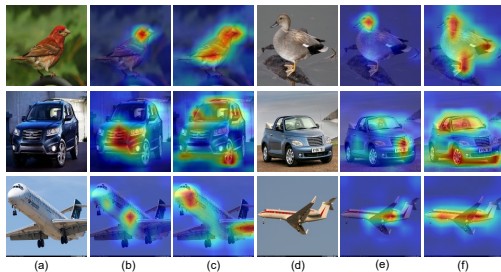

Figure 3: Visualization of the source of attributes. The top row shows the global view. The second and third rows show the multiple image patches after random cropping (local views). (a), (c) and (e) denote the input views. (b), (d) and (f) show the response maps.

Figure 4: Illustration of class activation maps produced by baseline and our VAPNet. (a) and (d) are the input images. (b) and (e) are the referred class activation maps by baseline. (c) and (f) denote the class activation maps provided by our VAPNet.

of the global view. Based on the above observation, we draw a conclusion that vision models can discover more semantic clues when replacing the input image with its local patches.

We exhibit the visualization results to demonstrate the influence of visual attributes. The referred visualizations of baseline and our model are shown in Fig. 4. It is shown that our model focuses on multiple local parts (*e.g.*, head, wings and abdomen, etc.) instead of the fixed part predicted by baseline, (*e.g.*, head of birds, front of cars, and middle of aircraft, etc). This verifies that using visual attributes learnt from known categories reasonably is beneficial for describing novel categories, thus improving the retrieval performance under open-set scenarios. More importantly, as shown in Fig. 4 (c) and (f), two sub-figures in the same row can roughly correspond to certain kinds of attributes of the fine-grained objects, *e.g.*, "wings of birds", "tires of cars", "head or wing of planes", etc. The results reflect that the activation of objects parts is apparently attribute-related and contains the visual discrepancies among unknown categories accordingly, which could provide a clear explanation of the success in retrieving unknown categories.

## 5 Conclusion

In this paper, we propose a novel Visual Attribution Parameterization Network (VAPNet) to handle unknown categories using visual attributes learnt from known instances in open-set fine-grained retrieval tasks. VAPNet focuses on distilling visual attributes from semantic clues presented in objects and utilizing these attributes as supervisory signals to tune the retrieval model. In this way, we could transform the retrieval model trained by image-level supervisions from category semantic extraction to attribute modeling, and precisely represent unknown categories based on its parameters supervised by visual attributes. Therefore, VAPNet successfully alleviates the problem behind facing instances from unseen novel categories. Last but not the least, the overall retrieval pipeline is simple and flexible. Extensive experiments demonstrate that our method outperforms the state-of-the-art methods by a significant margin, indicating the effectiveness of attribute modelling on facing unknown categories.

**Limitations & Broader Impacts:** By introducing VAPNet, we aim to extract visual attributes from seen classes without relying on attribute annotations to differentiate unseen classes. This innovation has the potential to greatly impact open-domain tasks. In particular, annotating a large number of attributes for unseen categories in open-domain tasks can be a costly and time-consuming endeavor. By enabling the model to automatically capture knowledge about unseen classes, our approach reduces the reliance on attribute annotations, resulting in decreased manual labeling costs. Furthermore, our approach exhibits improved adaptability to data from domains resembling the training set, such as natural images or medical images. This heightened adaptability contributes to stronger generalization capabilities, allowing the model to perform well in real-world scenarios. Ultimately, our solution has the potential to propel the advancement of open-domain tasks and facilitate their practical applications.

## Acknowledgements

This work is supported in part by the National Natural Science Foundation of China (NSFC) under Grants (No.61976038 and No.61932020), and The Taishan Scholar Program of Shandong Province (tstp20221128).

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
