# OpenReview forum: "Learning to Parameterize Visual Attributes for Open-set Fine-grained Retrieval"
_NeurIPS.cc/2023/Conference — NeurIPS 2023 poster_

### Official Review · Reviewer_Xapw · 2023-07-03

**Soundness:** 3 good
**Presentation:** 2 fair
**Contribution:** 3 good
**Rating:** 6
**Confidence:** 3

**Summary:**

The work proposes a new approach for open-set fine-grained retrieval. Their main contribution is the new state-of-the-art method termed VAPNet, which exploits objects’ attributes to discriminate between known and unknown objects. Due to the absence of attribute annotations, their training loss includes a pipeline to extract and refine attributes in an unsupervised fashion.

**Strengths:**

1. The authors’ idea to exploit visual attributes to discriminate between known and unknown classes is novel for the task.
2. The approach to learn such attributes in an unsupervised fashion and in the open-set scenario is quite interesting.
3. The quality of the manuscript is good.
4. Their evaluation protocol makes sense and their method achieves the best results across three datasets when compared with related works.

**Weaknesses:**

1. Section 3 is hard to follow, mainly the architectural choices. There are many components in the method. The authors could refer to Figure (2) whenever introducing some components to guide the reader’s understanding.
2. The differences in the pipeline between training and evaluation are not stated explicitly. This could help in understanding the overall method better.
3. Figure (2) could be improved to better convey the method (the current image is too crowded). From the text and the image it may be hard to understand whether the network represented in the retrieval module and the attribute exploration modules are the same or not. The retrieval embeddings returned by the pooling operations seems superfluous in the image. It may be best to either disentangle the components, or increase abstraction of some components. Moreover, the caption is too short and conveys no information. Ideally, the figure should be able to convey all that is required to understand the method without additional context.
4. This is a minor suggestion. Some section names are misleading or could be improved. For instance, many titles contain the word “attribute” to the point that its presence means little. Moreover, the title for Section 3.4 is misleading and it may be best to rename it.


**Questions:**

please refer to Weakness section

**Limitations:**

1. No limitation was reported. It may be compute intensive to extract multiple patches from the same image or for the overall training pipeline. Moreover, it could be that the evaluation process is slower due to the additional components introduced. It may be important to clearly explain any limitation to the work.
2. Same for broader impact. It should be stated

---

> ### Author Rebuttal · Authors · 2023-08-08
>
> Thank you for your thoughtful comments! Below, we discuss each of the reviewer's concerns, and explain how we plan to address them in the revised version of our manuscript.
>
> **W1**: Thank you for your valuable feedback.
> We appreciate your suggestion to refer to Figure (2) whenever introducing components in order to guide the reader's understanding. We agree that this will enhance the coherence of our narrative and make it easier for readers to follow along. We will make sure to incorporate these references in the revised version of our paper.
>
> Thank you once again for your insightful comments, which will help us improve the clarity and readability of our work.
>
> **W2**: Thank you for your insightful feedback. In the revised version, we will provide a more detailed explanation of the differences in the pipeline between training and evaluation. Specifically, our VAPNet framework comprises a backbone network, an attribute exploration module, and an attribute parameterization module. During the training phase, all these components are utilized to explore and parameterize visual attributes. However, during the evaluation phase, we only employ the backbone network to extract retrieval embeddings. By clearly highlighting these differences, we hope to enhance the overall understanding of our proposed method.
>
> **W3**: We sincerely thank the reviewer for the insightful feedback on Figure 2. Your suggestions are valuable, and we recognize the need for refining the figure to better represent our method. Aligning with your advice, we will present an improved version of Figure 2, which will hopefully address these concerns more effectively.
>
> **W4**: Thank you for bringing up the issue of section names. To enhance the clarity of our method description, we have decided to remove sub-titles such as attribute exploration, attribute sampling, and attribute parameterization constraint from the method section. This modification allows for a more streamlined and concise presentation of our approach. Additionally, we agree that the title for Section 3.4 is misleading and will rename it as "Loss Function" to better reflect its content. Thank you again for your valuable suggestion.
>
> **L1**: Thank you for your insightful comments. We appreciate your suggestions regarding the limitations of our work. As you correctly pointed out, the extraction of multiple patches during training can indeed introduce additional computational overhead. This approach is necessary to capture diverse visual attributes, which are then used as supervisory signals for fine-tuning the backbone network. While this does require computational resources during training, we believe that the benefits outweigh the costs, especially considering the expense of labeling attribute annotations.
>
> During the evaluation phase, it is important to note that our VAPNet does not utilize the designed components or extract multiple patches. As a result, the evaluation process is not slower than the baseline. To provide a comprehensive analysis of complexity, we have reviewed previous test logs and compared retrieval embedding extraction times and model parameters. The results are summarized in the table below:
>
> Method | Parameters | Time | Recall@1
> ----------|---------------|--------|-----------
> Baseline| 23.50M | 21.7ms | 69.5%
> Our VAPNet| 24.55M|21.7ms | 76.2%
>
> As observed, our proposed VAPNet with AAM and APM modules achieves a performance improvement of 6.7% while only adding an additional 1.05M parameters compared to the baseline. Furthermore, since the AAM and APM modules are only utilized during training, the retrieval embedding extraction time during testing remains the same as that of the baseline. Therefore, the increase in algorithm complexity is minimal and is considered acceptable.
>
> **L2**: We appreciate your feedback regarding the broader impact of our work.
> By introducing VAPNet, we aim to extract visual attributes from seen classes without relying on attribute annotations to differentiate unseen classes. This innovation has the potential to greatly impact open-domain tasks.
> In particular, annotating a large number of attributes for unseen categories in open-domain tasks can be a costly and time-consuming endeavor. By enabling the model to automatically capture knowledge about unseen classes, our approach reduces the reliance on attribute annotations, resulting in decreased manual labeling costs.
> Furthermore, our approach exhibits improved adaptability to data from domains resembling the training set, such as natural images or medical images. This heightened adaptability contributes to stronger generalization capabilities, allowing the model to perform well in real-world scenarios.
> Ultimately, our solution has the potential to propel the advancement of open-domain tasks and facilitate their practical applications. Thank you for bringing this to our attention.

---

> ### Comment · Reviewer_Xapw · 2023-08-16
> **Thanks for the response**
>
> The response clarified my comments, and I appreciate the authors' efforts in responding other reviewers too.
> In general, I think the method is interesting and comparison is convincing, please try to also make a clear presentation of it in your final version.

---

> > ### Author Response · Authors · 2023-08-16
> > **Response to Reviewer Xapw**
> >
> > Thank you for your comments. We appreciate your positive evaluation of our response and the overall review of our paper. We have taken note of your suggestions and will make an effort to improve the presentation of our work in our final version. Once again, we would like to express our gratitude for your valuable feedback and for contributing to the improvement of our manuscript.

---

### Official Review · Reviewer_uknX · 2023-07-06

**Soundness:** 2 fair
**Presentation:** 2 fair
**Contribution:** 2 fair
**Rating:** 5
**Confidence:** 3

**Summary:**

This paper proposes VAPNet to learn visual attributes from known fine-grained categories and parameterize them into final open-set retrieval. To learn visual attributes without attribute annotation, VAPNet explicitly attains some semantics with rich details via making use of local image patches and distills the visual attributes from these discovered semantics. Then, it incorporates the online refinement of these visual attributes into the training process to iteratively improve them and simultaneously regards these attributes as supervisory signals to tune the retrieval models. Experimental results on on open-set fine-grained retrieval benchmarks demonstrate improved performance compared to existing methods.
However, I have some concerns about the motivation and the methodology employed. Please refer to the detailed comments below.

**Strengths:**

- It is commendable to introduce the visual attributes to open-set fine-grained retrieval.
- This paper is well-written and easy to follow.

**Weaknesses:**

- The author do not explain the necessity of introducing visual attributes into open-set fine-grained retrieval.
- In line 149, the attribute exploration module attain semantic clues of input object via randomly cropping local patches from the input image, but is hard to guarantee the complete coverage of all discriminative local regions of the input image by random cropping.
- In addition to local attributes, visual attributes can also contain global attributes, such as shape. The design of attribute exploration module only focuses on local visual attributes while ignoring global visual attributes.
- The artificially defined attributes correspond to different visual features and can be decoupled from each other. While the proposed method does not further constrain the relationship between visual attributes(i.e. Eq.(6) the attribute parameterization constraint Lc), and the extracted visual attributes cannot be guaranteed to have the characteristics of this aspect. It is more like to enhance the local representation ability of the model.
- There are some errors in the details of the paper, such as Eq.(6) in line 322, which actually corresponds to Eq.(5).

**Questions:**

- The corresponding relationship between local features and different patch of global features in line 191 is not clearly explained.
- The global view in Figure 3(top row(b)(d)(f)) shows that the model only pays attention to some local parts, which is quite different from the VAPNet shown in Figure 4(top row(c)(f)), which focuses on more local parts.
- Does the randomly cropped size have an effect on the model, and what is the relationship between the cropped size, the Patch number M, and visual attribute dimension k.

---

> ### Author Rebuttal · Authors · 2023-08-08
>
> **W1**: Thank you for your valuable feedback. In open-set fine-grained retrieval, the model is required to learn embedding from the seen classes and then to be capable of utilizing the learned knowledge to distinguish the unseen classes. Existing approaches commonly employ class supervisory signals to guide the model in capturing discriminative details that are useful for identifying the seen classes. However, these approaches tend to focus solely on capturing discriminative concepts of known categories from unknown instances, which can make it challenging to identify unknown categories.
>
> A noteworthy observation in the context of fine-grained objects is that visually similar objects from different subcategories often share common visual attributes. Therefore, an unknown instance can be described comprehensively using a range of visual attributes, which can be discovered across multiple known categories. Leveraging this insight, we can represent the unknown classes using visual attributes that have been discovered from the seen instances. By combining these attributes, we can effectively capture the subtle differences between the unseen classes, thus mitigating the challenges associated with open-set scenarios.
>
> **W2**: Thanks for your nice comment. Random cropping cannot always guarantee the completeness of all the parts which are smaller than the size of the patch. Although there could exist some parts which are smaller than the patch size, those still have chances of getting split. However, this should not be a concern for model training, since we adopt random cropping which is a standard data augmentation strategy before the random cropping. This strategy leads to the result that patches are different compared with those of previous iterations. Small discriminative parts, which are split at this iteration due to the random cropping, will not be always split in other iterations. This variability in patches brings an additional advantage when dealing with occluded visual attributes. It improves the generalization ability of our model, allowing it to better handle occlusions and enhancing its overall performance.
>
> **W3**: VAPNet successfully captures global attributes with a high probability. We achieve this by integrating large-scale patches that cover 1/4 of the original image within our attribute exploration module. When small objects lie within these patches, VAPNet can directly capture their global attributes. However, it is important to note that these attributes serve as supervisory signals rather than object descriptors. Consequently, VAPNet transforms these global attributes into parameters. This allows us to effectively capture the global attributes of other large-scale objects using the parameters fine-tuned by global attributes produced those small objects.
>
> **W4**: This is due to the lack of attribute annotations, which makes it challenging to establish explicit relationships between attributes and guarantee specific characteristics for attributes.
>
> However, these limitations do not hinder the exploration of visual attributes for describing unknown classes. VAPNet is designed to capture diverse visual attributes and convert them into parameters for retrieval models. When provided with a local view, it can translate it into a set of attributes. Although this attribute set is not directly used for object description, it serves as supervisory signals to fine-tune the retrieval model's parameters. This process allows the retrieval model to automatically disentangle and store the attributes in its parameters. Thus, the model can produce precise attributes to describe diverse visual content and effectively capture the discriminative differences among unknown classes.
>
> As you correctly pointed out, VAPNet significantly enhances the local representation of the model. With this enhancement, VAPNet can accurately translate an input object into a set of visual attributes, leveraging its exceptional local perception.
>
> **W5**: Thanks for your valuable corrections. We will rectify this error in the revised version and meticulously refine our paper.
>
> **Q1**: Each local view includes its associated position information $(V_w, V_h, V_x, V_y)$, where $(V_w, V_h)$ represents the width and height of the local view, and $(V_x, V_y)$ denotes the center location of the local view. To extract local features from global features, we leverage the coordinate information of local views and employ the RoI Align operation proposed in Mask RCNN (He et al., ICCV 2017).
>
> **Q2**: The global view depicted in Fig. 3 is generated by the baseline network, rather than VAPNet. In contrast to Fig. 3, Fig. 4 is generated by VAPNet. The baseline network, which is supervised by class signals, tends to selectively learn partial regions that are easier to reduce the current training empirical risk for the seen categories. Consequently, it focuses more on specific local regions rather than capturing comprehensive details and information from all sides.
>
> **Q3**: The size of the randomly cropped patches does indeed have an impact on the performance. As the size of discriminative details may vary, it is important for the patch size to be flexible. Thus, we define four scales of patches to cover a wide range of detail scale. However, it is challenging to ensure that all scales of visual content are adequately covered solely through randomly cropped sizes.
>
> It is important to note that the cropped size, the number of patches (M), and the dimension of visual attributes (k) are independent of each other. Specifically, VAPNet aims to collect diverse visual attributes and transform them into model parameters, rather than utilizing them as the final representation. Hence, the number of patches is not a critical factor. Additionally, the size of patches only determines the richness of visual semantics, while the dimension of visual attributes indicates the complexity of these attributes. Therefore, they are not unrelated.

---

> > ### Comment · Reviewer_uknX · 2023-08-21
> >
> > Thanks to authors' reply. My main concerns are addressed, I update my rating: 4->5.

---

> > > ### Author Response · Authors · 2023-08-21
> > > **Response to Reviewer uknX**
> > >
> > > Thanks for raising your score! We’re very encouraged that our rebuttal basically addressed your concerns and appreciate your support for the paper's acceptance.

---

### Official Review · Reviewer_yK4j · 2023-07-07

**Soundness:** 3 good
**Presentation:** 3 good
**Contribution:** 3 good
**Rating:** 5
**Confidence:** 2

**Summary:**

Firstly, they introduce a novel approach to address the problem of open-set fine-grained retrieval settings. They transform the retrieval model, which is typically trained using image-level supervisions for category semantic prediction, into attribute modeling. This transformation helps alleviate the challenges posed by open-set fine-grained retrieval.

Secondly, they propose a Visual Attribution Parameterization Network (VAPNet) that distills visual attributes from various semantics observed in seen fine-grained objects. These attributes are then transcribed into parameters within the retrieval model. This parameterization allows for the precise representation of unknown categories based on their transformed parameters derived from visual attributes.

The authors conduct extensive experiments to evaluate their method's performance on open-set fine-grained retrieval tasks. The results demonstrate that their proposed approach, VAPNet, brings significant benefits. It achieves an average accuracy gain of 8.6% compared to the recent state-of-the-art work on three open-set fine-grained retrieval benchmarks.

**Strengths:**

The novelty presented in this paper revolves around the proposed Visual Attribution Parameterization Network (VAPNet) and its application in open-set fine-grained retrieval tasks.

The authors introduce VAPNet as a novel approach to handle unknown categories in such tasks. VAPNet focuses on distilling visual attributes from semantic clues observed in known instances. These visual attributes serve as supervisory signals to fine-tune the retrieval model. By doing so, the authors transform the retrieval model, originally trained with image-level supervisions for category semantic extraction, into attribute modeling. This transformation enables precise representation of unknown categories based on parameters supervised by visual attributes. As a result, VAPNet effectively addresses the challenges associated with encountering instances from unseen novel categories.

Furthermore, the authors highlight the simplicity and flexibility of the overall retrieval pipeline enabled by VAPNet. They emphasize that the proposed method surpasses state-of-the-art approaches by a significant margin, demonstrating the effectiveness of attribute modeling when dealing with unknown categories in fine-grained retrieval tasks.

**Weaknesses:**

1.	The framework generally follows some contrastive learning works. The input pair – image and patch, can be regarded as two views in the contrastive learning, which has been used in RegionCL[1]. Additionally, using KL divergence as objectives for distillation has been proposed in RepDistiller [2].

2.	The authors claim to solve the task of open-set fine-grained retrieval. However, the experiments on open-set image retrieval are missing in the paper.

3.	Why most of SOTA work reported in Table 2 are proposed for deep metric learning instead of the image retrieval? Additionally, why some benchmark datasets for image retrieval, such as CIFAR, COCO, Oxford, are not used here?

4.	There are lots of confusing concepts in the manuscript. For example, what does “c” and “k” in “… weight matrix W_L ∈ R^c x k …” in Line 179 and Line 194 refer to? What does “sampler” in L189 refer to? What does “L_s” in Eq. (8) refer to?

5.	There are some typo errors. For example, “A” in Eq. (2) should be in bold.

[1] Xu, Yufei, et al. "Regioncl: exploring contrastive region pairs for self-supervised representation learning." ECCV 2022.

[2] Yonglong Tian, et al. Contrastive Representation Distillation, ICLR 2020.

**Questions:**

Please reply the concerns in the weakness section during rebuttal.

**Limitations:**

Limitations are not discussed in this paper.

---

> ### Author Rebuttal · Authors · 2023-08-08
>
> Thank you for your thoughtful comments! Below, we discuss each of the reviewer's concerns, and explain how we plan to address them in the revised version of our manuscript.
>
> **W1**: Thank you for your insightful comments. I appreciate your corrections and would like to provide further clarification regarding the similarities and differences between our VAPNet framework and existing contrastive learning and knowledge distillation approaches.
>
> Indeed, our VAPNet framework shares some similarities with contrastive learning schemes, such as RegionCL [1]. The input pairs, consisting of an image and a patch, can be seen as two views in contrastive learning. However, the goals of our VAPNet differ from those of self-supervised learning schemes like RegionCL. While self-supervised learning aims to learn consistent probability distributions by attracting positive pairs and repelling negative pairs, our VAPNet utilizes local and global views to generate attribute pairs as supervisory signals. These attribute pairs are then used to transform the retrieval model from category prediction to attribute modeling.
>
> Similarly, you correctly mentioned that KL divergence is commonly used in knowledge distillation, as seen in techniques like RepDistiller [2]. However, the purpose of knowledge distillation is to make the output of a student network mimic that of a teacher network by constraining their probability distributions using KL divergence. In contrast, in our VAPNet, we employ KL divergence to ensure the distribution consistency between attribute pairs generated by local and global views within a single network. This allows us to iteratively refine the visual attributes during training and simultaneously use them as supervisory signals to fine-tune the retrieval models, resulting in effective attribute parameterization.
>
> Although input pairs and KL divergence have been widely used in various contexts, our VAPNet effectively combines them within a joint network and leverages their collaboration to transform the retrieval model from category prediction to attribute modeling. This unique integration makes our VAPNet specifically designed for open-set tasks, enabling the effective utilization of knowledge learned from seen categories to identify unseen categories. These characteristics offer valuable insights and practical applications for real-world scenarios.
>
> Thank you for your thoughtful feedback, and we will make sure to incorporate these additional explanations and clarifications in the revised version of our paper.
>
> **W2**: Thank you for bringing up this concern. We have indeed conducted experiments to evaluate the generalization ability of our VAPNet in open-set scenarios. As mentioned in Section 4.1, we have selected the CUB-200-2011, Stanford Cars, and FGVC Aircraft datasets, which are commonly used in open-set fine-grained retrieval tasks. These datasets are split into seen categories for training and unseen categories for evaluation, simulating real-world scenarios where new classes can emerge.
>
> In the experiments section, we report all the results under the scenario where our VAPNet is trained on the seen classes but tested on the unseen classes. This setup allows us to assess the performance of our method in handling unknown or novel query images, which is a key aspect of open-set retrieval.
>
> **W3**: Thank you for your question and feedback. In Table 2, we present the results for retrieving unseen classes. Metric learning is designed to quantify the similarity between two images independent of their class information. As a result, metric learning can be trained on seen classes and utilized for retrieving unknown classes. However, traditional image retrieval methods often suffer from being trapped in the discriminative knowledge of seen classes and may struggle to handle unseen subcategories effectively. Therefore, most of the state-of-the-art works reported in Table 2 are proposed for deep metric learning rather than image retrieval.
>
> Regarding the choice of benchmark datasets, general image retrieval datasets such as CIFAR, COCO, and Oxford cover a wide range of object categories. and their settings usually share the same classes in both training and testing. However, in our experiments, we specifically focused on open-set settings to simulate real-world scenarios where new classes can emerge. Therefore, we choose fine-grained datasets containing visually similar classes, which can be easily split into seen categories for training and unseen categories for evaluation.
>
> **W4**: Thanks for your valuable corrections. In $W_L$, "c" denotes the channel number of input features, while "k" denotes the dimension of visual attributes. "Sampler" refers to the RoI align operations used to crop the local features from the global features based on the coordinates of local views. We apologize for mistakenly writing the auxiliary constraint "$L_a$" instead of $L_s$". In the revised version, we will provide a clear explanation of these potentially confusing concepts and rectify any errors.
>
> **W5**: Thank you for your valuable feedback and for pointing out the typo errors in our manuscript. We appreciate your careful review and have made the necessary corrections, including ensuring that the "A" in Eq. (2) is in bold.

---

> > ### Comment · Reviewer_yK4j · 2023-08-18
> >
> > I have read the comments from other reviewers (especially the three Borderline reject comments) and the rebuttal. I continue to be borderline toward this paper.

---

> > > ### Author Response · Authors · 2023-08-18
> > > **Response to Reviewer yK4j**
> > >
> > > Thank you, Reviewer yK4j, for taking the time to review our rebuttal.
> > >
> > > If you have any remaining concerns after the rebuttal, we would be happy to resolve them before the end of the Author-Reviewer Discussions.
> > >
> > > Thank you again for your suggestions and for reviewing our work.
> > >
> > > Sincerely, Authors

---

### Official Review · Reviewer_CR86 · 2023-07-10

**Soundness:** 2 fair
**Presentation:** 2 fair
**Contribution:** 2 fair
**Rating:** 5
**Confidence:** 3

**Summary:**

The paper addresses the problem of fine-grained image retrieval, where the model must identify subtle object attributes in order to distinguish between visually similar classes.
To achieve this, the paper proposes a Visual Attribute Parameterization Network to localize object attributes in images to enhance performance.
Specifically, the paper designs Attribute Exploration Module that extracts image features into local patches and projects them into attribute space.
Moreover, an Attribute Parameterization Module is introduced to iteratively refine visual features for fine-grained retrieval.
The paper conducts experiments on three datasets of CUB, Stanford Cars, and FGVC Aircraft for retrieval tasks.

**Strengths:**

+ The direction of fine-grained image retrieval without attribute annotation is interesting with impactful real-world applications.
+ The idea of detecting attributes by using local image patches is sensible and has been demonstrated by prior works to be effective for fine-grained classification [6,11]
+ The paper shows promising retrieval improvements on CUB and Cars datasets.

**Weaknesses:**

+ The claim that "we are the first to transform the retrieval model trained by image-level supervisions from category semantic prediction into attribute modeling" is not well supported. To be specific, many works approach the problem of fine-grained recognition without attribute annotations [A, B] which also leverage the ideas of attribute localization just like the proposed method. Thus, the main difference between the proposed method and SOTA is simply the retrieval tasks instead of classification tasks, which lack novelty.
+ The reviewer is not yet convinced by the significance of the experimental results, as the paper lacks strong comparisons with appropriate baselines such as [A, B] that are specifically designed to capture fine-grained attributes without annotation. Thus, it is unclear to the reviewer how effective the proposed method is compared to SOTA.
+ The experiment datasets are small in scale and might not be challenging enough for grained classification. Specifically, CUB, CAR, and Aircraft datasets might not have a diverse number of attributes per class. Thus, the reviewer has some doubts about the effectiveness of the proposed method. It would be more convincing if experiments on recent fine-grained datasets such as DeepFashion [19] or iNaturalist [C].

[A] Lin et al., Bilinear CNN Models for Fine-Grained Visual Recognition
[B] Ding et al., Selective Sparse Sampling for Fine-Grained Image Recognition
[C] Grant Van Horn et al., The iNaturalist Species Classification and Detection Dataset

**Questions:**

Please refer to the weakness section

**Limitations:**

Sufficiently addressed

---

> ### Author Rebuttal · Authors · 2023-08-08
>
> We thank the reviewer for the positive comments as well as constructive suggestions. Below, we discuss each of the reviewer's concerns, and explain how we plan to address them in the revised version of our manuscript.
>
> **W1**:  Thank you for raising this point and providing references to related works. We appreciate your feedback and apologize for any confusion caused by our previous claim.
>
> As you correctly pointed out, it's true that there have been previous works that approach fine-grained recognition without attribute annotations and leverage ideas of attribute localization. However, we believe that our claim of being the first to transform the retrieval model trained by image-level supervisions from category semantic prediction into attribute modeling for identifying unknown categories is still well-supported. The key distinction lies in the open-set fine-grained retrieval task, which poses the challenge of **retrieving unseen subcategories using knowledge learned from seen subcategories**. This is different from the closed-set learning setting typically used in fine-grained recognition. Therefore, there are few studies that explore visual attributes from seen classes without attribute annotations and utilize these attributes to identify unknown classes.
>
> Specifically, works like Bilinear CNN [A] and S3N [B] focus on capturing discriminative details and identifying seen subcategories. However, when directly applying these fine-grained recognition methods to the open-set fine-grained retrieval task, the models tend to **get stuck in the discriminative knowledge of seen subcategories** and may struggle to handle unseen subcategories effectively.
>
> In VAPNet, we aim to address this challenge by transforming the retrieval model trained by image-level supervision into attribute modeling. By dissecting various fine-grained object semantics and capturing numerous attributes from seen categories, we aim to improve performance in recognizing unseen subcategories.
>
> While prior fine-grained recognition studies have also extracted visual attributes without additional attribute annotations, their acquired attributes are specific to seen categories and may not be useful for recognizing unseen subcategories. In contrast, our VAPNet method aims to capture attributes from seen data that aid in recognizing unseen subcategories.
>
> We appreciate your feedback and will make the necessary adjustments to our claim to better reflect our contribution in the revised version of the manuscript. Thank you for bringing this to our attention, and we apologize for any confusion caused.
>
> **W2**: Thank you for your feedback and raising the concern about the significance of our experimental results. Although Bilinear CNN [A] and S3N [B] were originally devised for recognizing seen subcategories, using them directly in retrieval tasks for unseen classes may result in poor performance. Concretely, to provide a clearer comparison, we have included the recall@k results for Bilinear CNN and S3N, as well as our VAPNet method, in the table below:
>
> Method | Recall@1 | Recall@2| Recall@4 | Recall@8
> ----------|----------|----------|----------|----------
> Bilinear CNN | 67.4% | 78.7% | 86.3% | 91.2%
> S3N | 65.4%| 77.2%|85.6% | 90.7%
> Our VAPNet | 76.2% | 84.6%| 90.1% | 94.0%
>
> From the results, it is evident that using Bilinear CNN and S3N directly in the retrieval tasks for unseen subcategories leads to lower recall@k compared to our VAPNet method. This highlights the challenge of using feature extractors trained in closed-set scenarios with classification supervisions to detect distinguishing variations from unseen subcategories, which ultimately affects the retrieval performance.
>
> In contrast, our VAPNet method focuses on acquiring visual attributes instead of relying solely on discriminative cues. This enables us to better comprehend the unknown categories and accurately represent their distinguishing variations, leading to a significant improvement.
>
> **W3**: Thank you for your valuable suggestions. We understand your doubts and agree that conducting experiments on recent fine-grained datasets, such as DeepFashion [19] or iNaturalist [C], would provide a more convincing evaluation.
>
> In response to your suggestion, we have performed an additional experiment on the large-scale DeepFashion dataset to further validate the effectiveness of our proposed VAPNet. The DeepFashion dataset offers an open-set retrieval dataset, where the 3,997 classes are split into training and the remaining 3,985 classes are used for testing. However, due to time limitations, we will consider conducting experiments on iNaturalist in future work to further verify the effectiveness of VAPNet.
>
> To provide a clearer comparison, we have included the recall@k results for our VAPNet method, as well as the state-of-the-art methods CEP [4] and PNCA [29], in the table below:
>
> Method | Recall@1 | Recall@10| Recall@20 | Recall@30|Recall@40
> ----------|----------|----------|----------|----------|----------
> CEP [4] ECCV20 | 90.6% | 98.0% | 98.6% | 98.9% | 99.1%
> PNCA [29] ECCV20| 90.9% | 98.2% | 98.9% | 99.1% | 99.4%
> Our VAPNet | 93.9% | 98.7% | 99.1% | 99.4% | 99.6%
>
> From the results, it is evident that our VAPNet method achieves superior performance compared to other state-of-the-art (SOTA) methods, CEP and PNCA, on the large-scale DeepFashion dataset. By leveraging the visual attributes learned from known instances to identify category-specific discrepancies, our VAPNet demonstrates impressive generalization capabilities.
>
> We greatly appreciate your professional feedback, and we will incorporate these additional experiments in the revised version of the paper to further demonstrate the effectiveness and generalization ability of our proposed VAPNet method. Thank you for your valuable input.

---

### Official Review · Reviewer_C9Cw · 2023-07-27

**Soundness:** 3 good
**Presentation:** 3 good
**Contribution:** 3 good
**Rating:** 6
**Confidence:** 3

**Summary:**

This paper works on an open-set fine-grain retrieval task. To align novel categories and the basic categories into a unified space, this paper proposes a Visual Attribute Parameterization Network (VAPNet) to learn visual attributes from known categories and parameterize them into the retrieval model. Extensive experiments on open-set fine-grained retrieval datasets validate the superior performance of our VAPNet over existing solutions.

**Strengths:**

- The motivation of this paper is sufficient and reasonable. Adopting a series of visual attributes to represent an image makes it easier to embed unknown categories and known categories into a unified space.

- The experimental results show this paper has achieved SOTA performance across three datasets.


**Weaknesses:**

- The visual attributes seem to be a series of local discriminative patch-level features by contrastive learning. It is a very implicit manner to extract attributes. With the development of large language models (LLM), it is possible to represent visual attributes as a structural language description through LLM-driven multi-modal model. Please discuss it.

- Additional attributes extraction model may bring additional computation. A detailed computation and parameters number discussion should be included.

**Questions:**

See weaknesses.

**Limitations:**

No limitations discussion is provided in this paper.

---

> ### Author Rebuttal · Authors · 2023-08-08
>
> We would like to thank the reviewer for taking the time to read our paper and thanks for your valuable feedback.
>
> **W1** : Thank you for your insightful comments.  As highlighted by the reviewers, recent multimodal large models, such as CLIP (Radford et al., ICML 2021) and BLIP2 (Li et al., Arxiv 2023), have shown success in learning the mapping between visual content and language concepts by training on visual and language tasks together.
>
> However, directly representing the visual attributes extracted by VAPNet as structured language descriptions poses a challenge. For example, one of the main difficulties arises from the fact that VAPNet and the text encoder of CLIP operate in different representation spaces. This semantic gap between our VAPNet and CLIP model makes it challenging to determine the most suitable language description for the visual attributes based on similarity between attribute and text embeddings. Similarly, BLIP2 encounters a comparable challenge as CLIP. Due to the absence of a semantic association between our VAPNet and the large language model in BLIP2, we are unable to directly input the attribute information into the large language model to generate the language description of visual attributes.
>
> One possible solution is to feed the patches used for visual attribute extraction into the image encoder of CLIP or BLIP2. This approach can help mitigate the modality gap problem. However, it is important to note that CLIP and BLIP2 are trained on image-text pairs that describe general categories rather than fine-grained attributes. This raises challenges in representing fine-grained visual attributes and may lead to ambiguities in language descriptions.
>
> We appreciate the opportunity to discuss this forward-thinking question and eagerly look forward to further discussions. We welcome any guidance or input to improve our response.
>
> **W2**: Thank you for raising this important concern. We carefully analyze the computation and parameter numbers for our proposed VAPNet method and will include a detailed discussion in our revised manuscript.
>
> As shown in the table below, we compare the parameters, retrieval embedding extraction times, and recall@1 performance of our VAPNet method with a baseline model.
>
> Method | Parameters | Time | Recall@1
> ----------|---------------|--------|-----------
> Baseline| 23.50M | 21.7ms | 69.5%
> Our VAPNet| 24.55M|21.7ms | 76.2%
>
> From the comparison, we can observe that our VAPNet method achieves a recall@1 performance improvement of 6.7% while only adding an additional 1.05M parameters compared to the baseline model. It is important to note that the increase in parameters is relatively small in proportion to the overall model size. Additionally, during testing, the retrieval embedding extraction time remains the same as that of the baseline model, as the additional attribution exploration modules (AAM and APM) are only used during training.
>
> Based on these results, we believe that the increase in computation and model complexity introduced by our additional attribute exploration model is minimal and acceptable. We will add this discussion to our revised manuscript to provide a comprehensive analysis of the computation and parameter numbers.
>
> We sincerely appreciate the time and effort you have invested in carefully reviewing our paper. We eagerly await your response and value your insights and feedback.

---

> > ### Comment · Area_Chair_mu4e · 2023-08-18
> >
> > thanks, all responses have been read and will be taken into account

---

> > > ### Author Response · Authors · 2023-08-18
> > > **Response to Area Chair mu4e**
> > >
> > > Thanks a lot for taking the time.  We’re very encouraged for your response and we would like to express our gratitude to you for your support!
> > >
> > > We believe that our proposed VAPNet specifically designed for open-set tasks plays a significant role in advancing practical applications for real-world scenarios.
> > >
> > > We hope our responses address all of concerns. Please feel free to raise further questions or concerns after you read our rebuttal.
> > >
> > > Thank you very much!
> > > Authors

---

### Author Response · Authors · 2023-08-18
**Looking Forward to Your Feedback**

Dear reviewers,

We appreciate your insightful reviews and detailed comments. We hope our responses address your concerns. Please feel free to raise further questions or concerns after you read our rebuttal. We hope to catch and answer them during the discussion period.

Thanks,
Authors

---

### Decision · Program_Chairs · 2023-09-21

**Decision:**

Accept (poster)

**Comment:**

After the rebuttal stage, there are two weak accept, two borderline accept, and one borderline reject scores. The reviewers praise the task and find the method interesting and effective. None of the concerns/questions asked in the reviews are critical; many are clarification questions. The borderline reject reviewer had the lowest confidence, and did not respond to the rebuttal, but their questions (mostly clarification, e.g. about the task) seem well addressed. The authors should improve the presentation as discussed during the rebuttal phase.